# GradInit: Learning to Initialize Neural Networks for Stable and Efficient Training

**Chen Zhu**
University of Maryland
chenzhu@cs.umd.edu

**Renkun Ni**
University of Maryland
rn9zm@cs.umd.edu

**Zheng Xu**
Google Research
xuzheng@google.com

**Kezhi Kong**
University of Maryland
kong@cs.umd.edu

**W. Ronny Huang**
Google Research
wrh@google.com

**Tom Goldstein**
University of Maryland
tomg@cs.umd.edu

## Abstract

Innovations in neural architectures have fostered significant breakthroughs in language modeling and computer vision. Unfortunately, novel architectures often result in challenging hyper-parameter choices and training instability if the network parameters are not properly initialized. A number of architecture-specific initialization schemes have been proposed, but these schemes are not always portable to new architectures. This paper presents GradInit, an automated and architecture agnostic method for initializing neural networks. GradInit is based on a simple heuristic; the norm of each network layer is adjusted so that a single step of SGD or Adam with prescribed hyperparameters results in the smallest possible loss value. This adjustment is done by introducing a scalar multiplier variable in front of each parameter block, and then optimizing these variables using a simple numerical scheme. GradInit accelerates the convergence and test performance of many convolutional architectures, both with or without skip connections, and even without normalization layers. It also improves the stability of the original Transformer architecture for machine translation, enabling training it without learning rate warmup using either Adam or SGD under a wide range of learning rates and momentum coefficients. Code is available at https://github.com/zhuchen03/gradinit.

## 1 Introduction

The initialization of network parameters has a strong impact on the training stability and performance of deep neural networks. Initializations that prevent gradient explosion/vanishing in back propagation played a key role in early successes with feed-forward networks [1, 2]. Even with cleverly designed initialization rules, complex models with many layers or multiple branches can still suffer from instability. For example, the original Transformer model [3] does not converge without learning rate warmup using the default initialization [4–6]; RoBERTa [7] and GPT-3 [8] have to tune the $\beta_2$ parameter of Adam for stability when the batch size is large. Recent innovations have shown that architecture-specific initializations, which are carefully derived to maintain stability, can promote convergence without needing normalization layers [5, 9–12]. Unfortunately, the reliance on analytically derived initializations makes it difficult to realize the benefits of these methods when performing architecture search, training networks with branched or heterogeneous components, or proposing altogether new architectures.

In this work, we propose a simple method for learning the initialization of a network with any architecture. Typically, initialization schemes draw parameters independently from a zero-mean distribution, with the variance of each distribution set to pre-determined values depending on the

35th Conference on Neural Information Processing Systems (NeurIPS 2021).

dimensions of the layers [1, 2]. Rather than deriving a closed-form expression for the these distribution parameters, our method re-scales each random weight tensor (e.g. convolution kernels) directly by a learned scalar coefficient. This small set of coefficients is optimized to make the first step of a stochastic optimizer (e.g. SGD or Adam) as effective as possible at minimizing the training loss, while preventing the initial gradient norm from exploding. In addition, this process is designed to take into account the direction, step size, and stochasticity of the optimizer. Finally, after the variance has been learned for each parameter tensor, the random network parameters are re-scaled and optimization proceeds as normal. We empirically find that our methods can make the initialization fall into a smooth loss region, reduce the inter-sample gradient variance, and accelerates training.

Our proposed method, GradInit, is architecture agnostic, and works with both Adam and SGD optimizers. In the vision domain, we show it accelerates the convergence and test performance of a variety of deep architectures, from the vanilla feed-forward VGG net to ResNet, with or without Batch Normalization. It is efficient and scalable, finding good initializations using less than 1% of the total training time in our experiments, and it improves the initialization of ResNet-50 on ImageNet to obtain better final test accuracy. In the language domain, GradInit enables training the original Transformer model [3] using either Adam or SGD without learning rate warmup for machine translation, which is commonly acknowledged to be difficult [4, 13]. As an extreme example of the capabilities of GradInit, we use it to initialize and train a 1202-layer ResNet that achieves significantly higher test accuracy than ResNet-110, which other initialization methods have failed to achieve.

Finally, by visualizing the initial norms and gradient variances of the weights before and after GradInit is applied, we show that GradInit is a useful tool for identifying potential causes for instability at initialization, such as those imposed by normalization layers, and we summarize interesting scale patterns learned by GradInit that can be helpful for designing better initialization rules.

## 2   Related Work

Controlling the norms of network parameters at initialization has proven to be an effective approach for speeding up and stabilizing training. Glorot and Bengio [1] studied how the variance of features evolves with depth in feed-forward linear neural networks by assuming both activations and weight tensors are independent and identical random variables. They developed a technique in which the variance of each filter scales with its fan-in (the number of input neurons). This style of analysis was later generalized to the case of ReLU networks [2]. These two analyses are most effective for feed-forward networks without skip connections or normalization layers. Based on the orthogonal initialization scheme [14], Mishkin and Matas [15] proposed an iterative procedure to rescale the orthogonally initialized weights of each layer in feedforward networks so that the activations of that layer have unit variance. However, this method fails to prevent the blowup of activations with depth for ResNets [16]. Recently, Gurbuzbalaban and Hu [17] proposed initialization schemes such that the network can provably preserve any given moment of order $s \in (0, 2]$ for the output of each layer. The motivation is that the stochastic gradient updates can result in heavy-tailedness in the distribution of the network weights with a potentially infinite variance, but finite $s$-order moment [18]. Again, these initialization schemes can only be applied for feed-forward neural networks.

For more complex architectures, normalization layers [19, 20] and skip connections [21] stabilized training dynamics and improved the state-of-the-art. Similarly, learning rate warmup is a common trick for training large Transformers [3]. These methods make training tractable for some models, but do not eliminate the high initial gradient variance that destabilizes training when the network is deep [9–11] or when the normalization layers are not carefully positioned [4].

Several authors have proposed better initializations for networks with skip connections. This is often achieved by replacing the normalization layers with simpler scaling or bias operations, and scaling the weight matrices in each layer so that the variance of activations does not increase with depth [9–12]. Similar analysis has been applied to self attention in Transformers [5]. Without removing the normalization layers, it is possbile to stabilize the initial parameter updates by introducing carefully initialized learnable scale factors to the skip connections [6] or the residual branches [22]. However, such techniques are often restricted to one specific architecture such as ResNets.

Recently, Dauphin and Schoenholz [16] proposed a task-agnostic and automatic initialization method, MetaInit, for any neural network achitecture. MetaInit optimized the norms of weight tensors to minimize the "gradient quotient", which measures the effect of curvature near the initial parameters,

on minibatches of random Gaussian samples. However, as training data is usually accessible for most tasks of interest, it is simpler and potentially more efficient to use the training data for initialization. MetaInit also involves the gradient of a Hessian-vector product that requires computing a "gradient of the gradient" multiple times in tandem, which is very computationally intensive. Our proposed method distinguishes itself from MetaInit in the following ways: (i) Our method is more computationally efficient. MetaInit involves computing third-order derivatives, results in long computing times and high memory usage. The memory overhead of MetaInit is more of an issue for networks with normalization layers. For the relatively small-scale CIFAR-10 problem with batch size 64, MetaInit requires three GPUs (RTX 2080Ti), while the proposed GradInit needs just one. (ii) Our method takes the stochasticity of minibatches into consideration. MetaInit uses the local curvature evaluated on a single minibatch, which fails to capture the variance of the loss/gradient between two different stochastic minibatches. (iii) Our method considers the training dynamics of different optimization algorithms including the learning rate and the direction of the gradient step, and effectively handles different optimizers including SGD and Adam.

## 3 Method

We aim to develop an initialization scheme applicable to arbitrary network architectures. Since previous works [1, 2, 9, 16, 10, 12] have shown that the initial weight norms effectively control the initial gradient norm on average, our method rescales the randomly initialized weight matrices using learnable scale factors.[1]

Using a small number of gradient descent steps on these scale factors, the proposed GradInit method chooses the initialization scalars so that the loss after the first gradient step taken by a stochastic optimizer (SGD or Adam) is as low as possible. The process of learning initialization coefficients accounts for the chosen learning rate, optimizer, and other parameters. To prevent gradient explosion, our method enforces a constraint that the gradient norm is no larger than a constant $\gamma$.

Note that for scale-invariant weights, e.g., convolution kernels before BN layers, rescaling still changes their learning dynamics by changing their effective learning rate [23, 24]. Empirically, GradInit goes beyond simply preventing exploding or vanishing gradients; it also reduces the gradient variance, making the initialization fall into a smooth loss region with small gradient variance so that training is fast, see discussion about Figure 1 and comparisons in Figure 2.

### 3.1 Efficient Learning-based Initialization via Constrained Optimization

We begin by filling all the weight matrices $\{\boldsymbol{W}_1, \ldots, \boldsymbol{W}_M\}$ of the network with values drawn from independent zero-mean Gaussian distributions, except for the scales and biases of the normalization layers (if any), which are initialized to 1 and 0 respectively. During the initialization process, we keep $\{\boldsymbol{W}_1, \ldots, \boldsymbol{W}_M\}$ constant, but we multiply each $\boldsymbol{W}_i$ with a learnable non-negative scale factor $\alpha_i$ (initialized to 1). After initialization, we rescale the weights by the learned scale factors, and start training without the learnable scale factors just as normal. We use $\boldsymbol{m} = \{\alpha_1, \ldots, \alpha_M\}$ to denote the set of scale factors, and $\boldsymbol{\theta_m} = \{\alpha_1 \boldsymbol{W}_1, \ldots, \alpha_M \boldsymbol{W}_M\}$ is the set of rescaled weight matrices.

Let $L(S; \boldsymbol{\theta}) = \frac{1}{|S|} \sum_{x \in S} \ell(x; \boldsymbol{\theta})$ be the average loss of the model parameterized by $\boldsymbol{\theta}$ on a minibatch of samples $S$, where $|S|$ is the number of samples in the minibatch. We use $\boldsymbol{g}_{S,\boldsymbol{\theta}} = \nabla_{\boldsymbol{\theta}} L(S; \boldsymbol{\theta})$ as a shorthand for the gradient of $\boldsymbol{\theta}$. During standard training, this gradient is preprocessed/preconditioned by the optimization algorithm $\mathcal{A}$, and then used to update the network parameters. GradInit solves the following constrained optimization problem:

$$\begin{aligned}
\underset{\boldsymbol{m}}{\text{minimize}} \quad & L(\tilde{S}; \boldsymbol{\theta_m} - \eta \mathcal{A}[\boldsymbol{g}_{S,\boldsymbol{\theta_m}}]), \\
\text{subject to} \quad & \|\boldsymbol{g}_{S,\boldsymbol{\theta_m}}\|_{p_{\mathcal{A}}} \leq \gamma,
\end{aligned} \quad (1)$$

where $S$ and $\tilde{S}$ are two different minibatches, $\eta$ is a prescribed learning rate for the optimization algorithm $\mathcal{A}$, $p_{\mathcal{A}}$ is the $\ell_p$-norm associated with $\mathcal{A}$, and $\gamma$ is the upper bound for the norm. For the first gradient step, Adam uses $\mathcal{A}[\boldsymbol{g}_{S,\boldsymbol{\theta_m}}] = \text{sign}(\boldsymbol{g}_{S,\boldsymbol{\theta_m}})$ [25], while SGD uses $\mathcal{A}[\boldsymbol{g}(S; \boldsymbol{\theta_m})] =$

---

[1]For convenience, we refer to weight vectors/matrices/tensors as "weight matrices", which includes the scale vectors of the normalization layers, the bias vectors, the weight matrices of the fully connected layers, and the convolution kernels.

$\gamma \boldsymbol{g}(S; \boldsymbol{\theta_m}) / \|\boldsymbol{g}(S; \boldsymbol{\theta_m})\|_2$. We show how to choose $\gamma$ and $p_{\mathcal{A}}$ without tuning in Section 3.3. We discuss the formulation of this problem and how to solve it below.

## 3.2 Solving the Constrained Problem

The problem (1) is solved using a stochastic gradient descent method in which we sample new mini-batches on each iteration. Since the proposed method uses gradient updates to compute the initialization, we dub it *GradInit*. We propose a simple solver to optimize objective (1) in Algorithm 1. A key feature of our method is that is makes a simple approximation: after $\boldsymbol{g}_{S,\boldsymbol{\theta_m}}$ is computed on the forward pass of an iteration, we treat $\mathcal{A}[\boldsymbol{g}_{S,\boldsymbol{\theta_m}}]$ as a constant and do not back-propagate through $\mathcal{A}[\boldsymbol{g}_{S,\boldsymbol{\theta_m}}]$ on the backward pass. We make this choice to keep computing costs low, and because it is not possible to back-propagate through the non-differentiable sign function for Adam.

---

**Algorithm 1** *GradInit* for learning the initialization of neural networks.

---

1: **Input:** Target optimization algorithm $\mathcal{A}$ and learning rate $\eta$ for model training, initial model parameters $\boldsymbol{\theta}_0$, learning rate $\tau$ of the GradInit scales $\boldsymbol{m}$, total iterations $T$, upper bound of the gradient $\gamma$, lower bound for the initialization scalars $\underline{\alpha} = 0.01$.
2:    $\boldsymbol{m}_1 \leftarrow \boldsymbol{1}$
3: **for** $t = 1$ **to** $T$ **do**
4:      Sample $S_t$ from training set.
5:      $L_t \leftarrow \frac{1}{|S_t|} \sum_{x_k \in S_t} \ell(x_k; \boldsymbol{\theta_{m_t}})$, $\boldsymbol{g}_t \leftarrow \nabla_{\boldsymbol{\theta}} L_t$
6:      **if** $\|\boldsymbol{g}_t\|_{p_{\mathcal{A}}} > \gamma$ **then**
7:        $\boldsymbol{m}_{t+1} \leftarrow \boldsymbol{m}_t - \tau \nabla_{\boldsymbol{m}_t} \|\boldsymbol{g}_t\|_{p_{\mathcal{A}}}$
8:      **else**
9:        Sample $\tilde{S}_t$ from training set.
10:        $\tilde{L}_{t+1} \leftarrow \frac{1}{|\tilde{S}_t|} \sum_{x_k \in \tilde{S}_t} \ell(x_k; \boldsymbol{\theta_{m_t}} - \eta \mathcal{A}[\boldsymbol{g}_t])$
11:        $\boldsymbol{m}_{t+1} \leftarrow \boldsymbol{m}_t - \tau \nabla_{\boldsymbol{m}_t} \tilde{L}_{t+1}$
12:      Clamp $\boldsymbol{m}_{t+1}$ using $\underline{\alpha}$

---

To enforce the constraint in (1), we test whether the constraint is satisfied after computing $\boldsymbol{g}(S; \boldsymbol{\theta_m})$. If not, we take a gradient descent step to minimize $\|\boldsymbol{g}(S; \boldsymbol{\theta_m})\|_{p_{\mathcal{A}}}$, which involves computing second order derivatives. If the constraint is satisfied, then we instead compute a gradient descent step for the loss. In addition, we set a lower bound $\underline{\alpha} = 0.01$ for all $\alpha_i$. We find that this prevents scalars from landing on small values during minimization and keeps the GradInit optimizer stable. In our experiments, we find the only layer that ever hit this lower bound is the final FC layer on some networks (see the figures in Section 4.1). We find this procedure converges reliably within 2000 iterations for ImageNet, and fewer than 400 iterations for CIFAR-10, taking less than 1% of the total training time on both problems. We also find it works well to set the step size $\tau$ to values within the range between $10^{-3}$ and $10^{-1}$. During initialization, the gradient norm constraint is satisfied for the majority of iterations. The choice of $\gamma, p_{\mathcal{A}}$ will be discussed in Section 3.3.

**Stochasticity of mini-batching.** The objective in (1) uses two different mini-batches; $S$ is used to compute the gradient, and $\tilde{S}$ is used to compute the loss. Ideally, $S$ and $\tilde{S}$ should be independently sampled from the training set to capture the randomness of the stochastic optimizer. However, when the network has large initial gradient variance, the gradients on $S$ and $\tilde{S}$ usually differ a lot, and for $\tilde{S}$, the gradient update step $\boldsymbol{\theta_m} - \eta \mathcal{A}[\boldsymbol{g}_{S,\boldsymbol{\theta_m}}]$ becomes more similar to adding random perturbations to the parameters. We find our objective less effective at accelerating conver-

Table 1: Accuracies on CIFAR-10 using different overlapping ratios of $\tilde{S}$ and $S$ for GradInit.

| Model | $\frac{|\tilde{S} \cap S|}{|S|}$ | $Acc_1$ | $Acc_{best}$ |
|---|---|---|---|
| VGG-19 | 0 | $21.9 \pm 4.4$ | $94.5 \pm 0.1$ |
| w/o BN | 0.5 | $\mathbf{29.3 \pm 0.6}$ | $\mathbf{94.7 \pm 0.02}$ |
| (20.03 M) | 1 | $28.7 \pm 1.0$ | $94.5 \pm 0.1$ |

gence in this case, as shown by the first-epoch accuracy ($Acc_1$) in Table 1. On the other hand, the randomness is not captured if $S = \tilde{S}$, and we find empirically that $\boldsymbol{\theta_m}$ can exploit the loss by increasing the gradient norm and destabilize training in this case (see Table 8). Without excessive tuning, we find that we get more reliable behavior for different architectures when $\tilde{S}$ is a mixture of 50% samples from $S$ and 50% re-sampled training data, and use this setting by default unless otherwise stated.

## 3.3 Setting and Enforcing the Constraint

The constraint in (1) is included to prevent the network from minimizing the loss in a trivial way by blowing up the initial gradient. In other words, we want the optimizer to achieve small loss by choosing an effective search direction rather than by taking an extremely large step in a sub-optimal direction.

**Setting $p_{\mathcal{A}}$ and $\gamma$ through first-order approximation.** We show that $p_{\mathcal{A}}$ and $\gamma$ can be set easily with a rule of thumb and without a parameter search. From the first-order approximation, we expect the first gradient step to result in a change in the loss on $S$ as following:

$$L(S;\theta_{\boldsymbol{m}}-\eta\mathcal{A}[\boldsymbol{g}_{S,\theta_{\boldsymbol{m}}}])-L(S;\theta_{\boldsymbol{m}}) \approx -\eta\mathcal{A}[\boldsymbol{g}_{S,\theta_{\boldsymbol{m}}}]^T\boldsymbol{g}_{S,\boldsymbol{m}} = \begin{cases} -\eta\|\boldsymbol{g}_{S,\theta_{\boldsymbol{m}}}\|_2^2, & \text{if } \mathcal{A} \text{ is SGD,} \\ -\eta\|\boldsymbol{g}_{S,\theta_{\boldsymbol{m}}}\|_1, & \text{if } \mathcal{A} \text{ is Adam.} \end{cases} \quad (2)$$

To effectively bound the approximated change in Eq. 2, we choose $\ell_{p_{\mathcal{A}}}$ to be the $\ell_2$ and $\ell_1$ norm for SGD and Adam respectively, so when the constraint is satisfed, the maximum change in the loss, according to our local approximation, is $\eta\gamma^2$ for SGD and $\eta\gamma$ for Adam. We recommend setting $\gamma$ such that $\eta\gamma^2 = 0.1$ for SGD and $\eta\gamma = 0.1$ for Adam. According to the linear approximations, this limits the gradient magnitude so that the first step of SGD can decrease the loss by at most 0.1. This simple rule was used across all vision and language experiments.

**Why a constraint and not a penalty?** Instead of formulating GradInit as a constrained optimization, one can alternatively formulate it as minimizing the objective with a gradient penalty: $\underset{\boldsymbol{m}}{\text{minimize}} \quad L(\tilde{S};\theta_{\boldsymbol{m}} - \eta\mathcal{A}[\boldsymbol{g}_{S,\theta_{\boldsymbol{m}}}]) + \lambda\|\boldsymbol{g}_{S;\theta_{\boldsymbol{m}}}\|_{p_{\mathcal{A}}}$, where $\lambda > 0$ is the penalty strength.

The penalized objective has two drawbacks compared to the constrained one in Eq. 1. First, every gradient descent step on the penalized objective involves second-order gradients due to the gradient regularization, while the constrained form does not need second-order gradients when the constraint is satisfied. Second, it is difficult to choose a good $\lambda$ that works well for all architectures. By contrast, we set $\gamma$ by analyzing the first-order approximation mentioned above, and find the same $\gamma$ works well for different architectures. The results supporting these two points are given in Table 2.

Table 2: Time cost and accuracy (average of 4 runs) for running one epoch of regularization/constrained form of GradInit.

| Model | VGG-19 w/o BN | VGG-19 w/ BN | ResNet-110 w/o BN | ResNet-110 w/ BN |
|---|---|---|---|---|
| Time (s) | 82 vs. 56 | 100 vs. 62 | 169 vs. 103 | 269 vs. 195 |
| $\lambda = 10^{-4}$ | **32.3**, 94.6 | 10.6, 93.1 | 33.7, 93.9 | 32.4, 95.2 |
| $\lambda = 10^{-2}$ | 30.4, 94.5 | 10.4, 93.0 | **36.7**, 94.1 | 32.6, 95.3 |
| $\lambda = 1$ | 18.2, 74.7 | 38.5, 95.1 | 30.7, 94.2 | 36.5, 95.3 |
| $\gamma = 1$ | 29.3, **94.7** | **47.8**, **95.1** | 36.2, **94.6** | **38.2**, **95.4** |

## 4 Experiments

We evaluate GradInit on benchmark datasets for image classification and machine translation tasks. For image classification, five different architectures are evaluated for CIFAR10 [26], and ResNet-50 is evaluated for ImageNet [27]. For machine translation, we use GradInit to find good initializations for a Post-LN Transformer without any change to its original architecture on IWSLT-14 De-En [28]. We observe that the method can remove the necessity of any form of learning rate warmup for both Adam and SGD.

We conduct our experiments in PyTorch. We use the fairseq library for machine translation [29]. All the experiments on CIFAR-10 and IWSLT-14 DE-EN can run with one single NVIDIA RTX 2080 Ti GPU with 11GB of RAM.

GradInit first initializes the weights using Kaiming initialization [2] for all the Conv and FC layers for image classification. For machine translation, we use the default Xavier initialization [1]. We optimize the scale factors $\{\alpha_i\}$ with Adam [30] using the default momentum parameters.

### 4.1 Image Datasets with Various Architectures

The introduction of Batch Normalization (BN) [19] and skip connections makes it relatively easy to train common CNNs for image classification to achieve high accuracy. Despite this, we show that

when the network is very deep, the network is unstable even when both BN and skip connections are used, and GradInit can significantly improve the stability. The results on CIFAR-10 are given in Table 3 and results on ImageNet are given in Table 6.

### 4.1.1 Settings

**Architectures.** On CIFAR-10, we focus on the feedforward VGG net and the prevalent and powerful ResNet, with and without BN layers. For networks without BN, we use learnable biases in all layers. For ResNet, we additionally evaluate a deep 1202-layer version. We give results for other architectures (Wide ResNet, DenseNet) in Appendix E due to space limits. We compare with four different methods/settings: 1) Kaiming Initialization [2]; 2) First train the network for one epoch with a constant learning rate equal to the starting learning rate, labelled as "+1 epoch (Const. LR)" in Table 3; 3) First train the network for one epoch with a linear warmup learning rate, labbeled as "+1 epoch (Warmup)" in Table 3; 4) MetaInit [16].

On ImageNet, we use the ResNet-50 model [21]. We compare with Kaiming Initialization, FixUp initialization [9] and MetaInit. For the ResNet-50 without BN, we follow the architecture of FixUp for fair comparisons, but we still use the original Kaiming initialization as the starting point of GradInit.

**Hyperparameters.** We set $\mathcal{A}$ to SGD and $\eta = 0.1$ (the same as the base learning rate) for GradInit in all image classification experiments. On CIFAR-10, we train networks with a batch size of 128. We find MetaInit often takes 2 to 3 times as much memory as GradInit. We run GradInit or MetaInit for one epoch on the data, which takes less than 1% of the total training time. For GradInit, according to our analysis in Section 3.3, we fix the gradient norm constraint $\gamma = 1$ in all these experiments. Therefore, as in MetaInit, the only hyperparameter that needs to be tuned is the learning rate $\tau$ of the scale factors. We do a grid search on $\tau$ in the range $[10^{-3}, 10^{-1}]$, and report the results with the best average final test accuracy on 4 runs. After GradInit initialization, we use a learning rate of 0.1 and the cosine annealing learning rate schedule without restart [31] to train the model for 200 epochs, where the learning rate decays after each iteration and decays to 0 in the last iteration. Due to their high initial gradient variance (see Figure 6), we have applied gradient clipping (maximum norm is 1) to all non-BN networks so that they converge without GradInit under the same schedule.

On ImageNet, we train the ResNet-50 model for 90 epochs with a total batch size of 256 on 4 GPUs. Due to the difference in the library for training and the number of GPUs used, which affects the BN statistics, our baseline top-1 accuracy of ResNet-50 (w/ BN) on ImageNet is 0.79% lower than [32]. We use SGD with a starting learning rate of 0.1 and decay the learning rate by 10 after the 30th and 60th epoch. We provide additional details in Appendix A.

### 4.1.2 Results and Analysis

Table 3: First epoch ($Acc_1$) and best test accuracy over all epochs ($Acc_{best}$) for models on CIFAR-10. We report the mean and standard error of the test accuracies in 4 experiments with different random seeds. Best results in each group are in bold.

| Model (# Params) | | VGG-19 w/o BN (20.03M) | VGG-19 w/ BN (20.04M) | ResNet-110 w/o BN (1.72M) | ResNet-110 w/ BN (1.73M) | ResNet-1202 w/ BN (19.42M) |
|---|---|---|---|---|---|---|
| Kaiming | $Acc_1$ | $29.1 \pm 1.5$ | $12.6 \pm 0.6$ | $16.1 \pm 2.1$ | $23.2 \pm 0.9$ | $12.9 \pm 2.8$ |
| | $Acc_{best}$ | $94.5 \pm 0.1$ | $94.4 \pm 0.1$ | $94.2 \pm 0.1$ | $95.0 \pm 0.2$ | $94.4 \pm 0.6$ |
| +1 epoch (Const. LR) | $Acc_1$ | $37.2 \pm 1.1$ | $19.6 \pm 4.0$ | $21.0 \pm 3.8$ | $32.5 \pm 3.8$ | $12.6 \pm 2.8$ |
| | $Acc_{best}$ | $94.4 \pm 0.1$ | $94.5 \pm 0.1$ | $93.9 \pm 0.4$ | $94.7 \pm 0.3$ | $94.0 \pm 0.4$ |
| +1 epoch (Warmup) | $Acc_1$ | $37.4 \pm 1.2$ | $53.5 \pm 2.9$ | $19.8 \pm 0.5$ | $48.7 \pm 1.1$ | $28.1 \pm 1.3$ |
| | $Acc_{best}$ | $94.4 \pm 0.1$ | $94.7 \pm 0.1$ | $94.1 \pm 0.1$ | $95.1 \pm 0.1$ | $95.4 \pm 0.2$ |
| MetaInit | $Acc_1$ | $30.5 \pm 0.9$ | $35.1 \pm 0.6$ | $14.6 \pm 2.2$ | $29.0 \pm 1.5$ | $11.7 \pm 1.6$ |
| | $Acc_{best}$ | $94.6 \pm 0.1$ | $94.6 \pm 0.1$ | $94.2 \pm 0.1$ | $94.8 \pm 0.1$ | $95.0 \pm 0.5$ |
| GradInit | $Acc_1$ | $29.3 \pm 0.6$ | $47.8 \pm 1.8$ | $36.2 \pm 0.8$ | $38.2 \pm 0.9$ | $29.0 \pm 1.1$ |
| | $Acc_{best}$ | $\mathbf{94.7} \pm 0.1$ | $\mathbf{95.1} \pm 0.1$ | $\mathbf{94.6} \pm 0.1$ | $\mathbf{95.4} \pm 0.1$ | $\mathbf{96.2} \pm 0.1$ |

**GradInit further stabilizes feedforward nets with BN.** BN does stabilize VGG-19 and allows training without gradient clipping, but with an average first-epoch test accuracy of only 12.57 and an average final test accuracy lower than the version without BN (see Table 3), it does not seem to

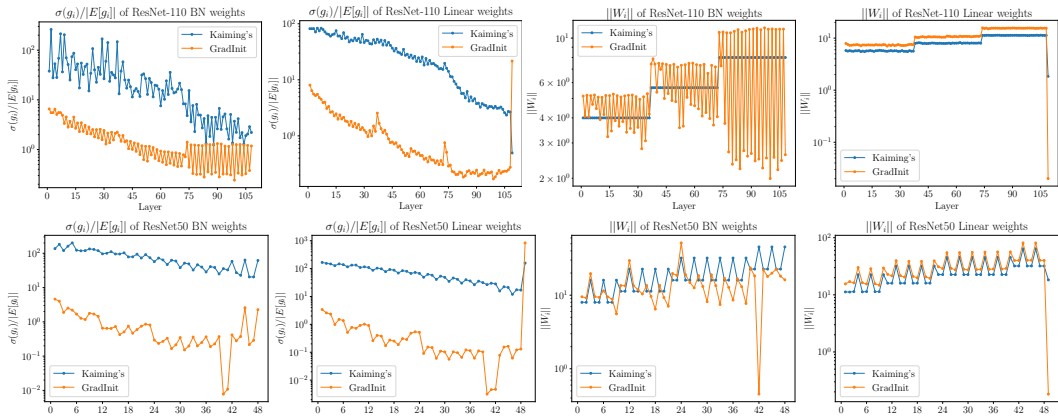

Figure 1: Top row: results of ResNet-110 on CIFAR-10. Bottom row: results of ResNet-50 on ImageNet. Left two columns: compare the relative cross-batch gradient variance on the training set for the BN and Conv/FC layers before and after GradInit. Right two columns: weight norms before and after GradInit. Ratio between points in the same layer reflects the scale factor. Note each of the residual blocks has 2 and 3 Conv and BN layers for the ResNet-110 and ResNet-50, respectively. The initial relative gradient variance are reduced for all layers except the final linear layer in both settings. The strategies are similar on two different datasets. Within each residual block, the last BN layer has the smallest scaling factors, and the scales of all Conv layers are surprisingly increased. Best viewed in color.

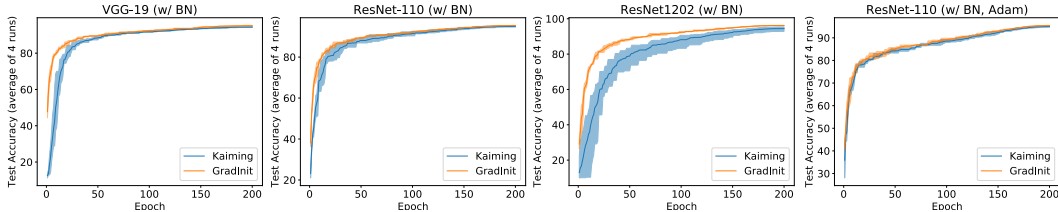

Figure 2: Comparing the convergence of Kaiming Initialization and GradInit on CIFAR-10, for models trained with SGD (left three) and Adam (right).

eliminate the instability of Kaiming initialization. As shown in Figure 4, its initial gradient variance is still relatively high compared with GradInit. BN could magnify the gradient variance when the variance of its input features (in the forward pass) is smaller than 1 (see Appendix C). GradInit reduces the gradient variance by 4 orders of magnitude compared to Kaiming initialization , resulting in significantly higher test accuracy after the first epoch (47.79% vs. 12.57%), which also has an impact on the final test accuracy (95.13% vs. 94.41%). The reduction in gradient variance is achieved mainly by scaling down the weights of the final FC layer and the last 2 BN layers, so that the variance of the activations is reduced in the forward pass. This learned behavior is consistent with the strategy of FixUp, where the final FC layer is initialized to 0. Another source of gradient variance reduction is achieved by *increasing* the weight norms of the remaining Conv and BN layers, so that the variance of the inputs to the BN layers is increased and the gradient magnifying effect of BN is alleviated in the backward pass. This reduced the ratio $\sigma(\boldsymbol{g}_1)/\sigma(\boldsymbol{g}_{16})$ from 204.9 to 164.8 for the Conv layers in Figure 4. By contrast, FixUp only reduces the weight norms, which may not always be the best solution for networks with normalization layers.

**Deep residual networks still need better initializations.** We also gain significant improvements from GradInit for ResNet-110 and ResNet-1202. In ResNets, the skip connections cause the variance of activations to accumulate as the ResNet goes deeper, even for the version with BN [10]. This issue is more significant when the ResNet scales to 1202 layers, from which we can see that with Kaiming initialization, the first-epoch accuracy of ResNet-1202 is quite low, and the final test accuracy is even worse than the shallower ResNet-110, matching the observations of He et al. [21]. Warmup is even more effective than MetaInit at accelerating the convergence and improving the final test accuracy of ResNet-1202, but GradInit still outperforms its final test accuracy by 0.8%, and the resulting ResNet-1202 finally achieved higher accuracy than ResNet-110.

The learned layer-wise rescaling patterns of GradInit are even more interesting for ResNets with BN. For ResNets with BN, recall that we have two Conv layers and two BN layers in each residual block. As shown in Figure 1, GradInit learns to *increase* the weight norms of all the linear layers except for

the final FC layer, instead of decreasing as for the case without BN (see Figure 6). A more unique pattern is the collaborative behavior of the BN weights, where the second BN in each residual block is usually scaled down while the first BN is always scaled up. In deeper layers, the joint effect of these two BN weights is to downscale the activations and reduce their variance in the forward pass, with a more significant reducing effect as the layers get deeper. Intuitively, the marginal utility of adding a new layer decreases with depth. Therefore, for deeper layers, GradInit learns to further downscale the residual branch, and prevents the variance from increasing too much in the forward pass. Inside each residual block, increasing the scale factors of the first BN helps to reduce the magnification effect of the second BN on the gradient; forcing the input activations to the second convolution to have variance larger than 1 ensures its variance after the following convolution layer does not go below 1, avoiding the magnification effect that the second BN has on the gradient variance. See Appendix C for more discussions about the magnifying effect.

Table 4: Comparing the results of GradInit with fixed BN scale parameters (Fix BN) and only rescale the BN parameters (Only BN).

| Model | Kaiming | | GradInit | | GradInit (Fix BN) | | GradInit (Only BN) | |
|---|---|---|---|---|---|---|---|---|
| | $Acc_0$ | $Acc_{best}$ | $Acc_0$ | $Acc_{best}$ | $Acc_0$ | $Acc_{best}$ | $Acc_0$ | $Acc_{best}$ |
| VGG-19 (w/ BN) | $12.6 \pm 0.6$ | $94.4 \pm 0.1$ | $47.8 \pm 1.8$ | $95.1 \pm 0.1$ | $13.1 \pm 0.9$ | $94.6 \pm 0.1$ | $14.4 \pm 2.1$ | $94.4 \pm 0.1$ |
| ResNet-110 (w/ BN) | $23.2 \pm 0.9$ | $95.0 \pm 0.2$ | $38.2 \pm 0.9$ | $95.4 \pm 0.1$ | $24.7 \pm 3.1$ | $94.7 \pm 0.3$ | $25.4 \pm 3.1$ | $94.6 \pm 0.3$ |

Table 5: Comparing the results with multiplying each weight matrix with a learnable scaler (Learning Scalars) on CIFAR10. The VGG-19 model is not able to converge unless we reduce the initial learning rate to 0.01, which obtained worse final accuracy. The ResNet-110 model's $Acc_0$ was 10% for 2 of the 4 runs.

| Model | Learning Scalars | | GradInit | |
|---|---|---|---|---|
| | $Acc_0$ | $Acc_{best}$ | $Acc_0$ | $Acc_{best}$ |
| VGG-19 (w/ BN, lr=0.1) | $10.0 \pm 0.0$ | $10.0 \pm 0.0$ | $47.8 \pm 1.8$ | $95.1 \pm 0.1$ |
| VGG-19 (w/ BN, lr=0.01) | $50.6 \pm 0.8$ | $93.4 \pm 0.1$ | - | - |
| ResNet-110 (w/ BN) | $21.5 \pm 6.9$ | $94.7 \pm 0.1$ | $38.2 \pm 0.9$ | $95.4 \pm 0.1$ |

**Generalizing to Adam.** Models in previous experiments are trained with SGD. We also consider the case when $\mathcal{A}$ is Adam and use AdamW [33] to train the ResNet-110 (w/ BN) model on CIFAR-10. Following [34], we use a cosine annealing learning rate schedule with initial learning rate $3 \times 10^{-3}$ and weight decay 0.2. For GradInit, we set $\gamma = 25$. The $Acc_1$ and $Acc_{best}$ of Kaiming initialization and GradInit are $(36.6 \pm 4.7, 94.9 \pm 0.1)$ and $(40.2 \pm 0.2, 95.3 \pm 0.1)$, respectively. We also show the per-epoch test accuracy in Figure 2.

**The importance of rescaling BN layers.** The scale parameters of BN layers usually controls the variance of activations and gradients in the forward and backward passes, while the linear layers right before the BN layers are scale-invariant. Although changing the magnitudes of the scale-invariant layers affect their learning dynamics [23, 24], we find it important for GradInit to rescale both BN and other linear layers, as shown in Table 4.

**The importance of GradInit's objective.** GradInit is designed to rescale the layers to solve the constrained optimization problem in Eq. 1. Simply letting the model to learn to rescale the layers cannot improve the results, and sometimes further causes instability, as shown in Table 5. We hypothesize that the bad results with VGG are due to a mismatch between the scales/norms of the gradients of the scalars and the weights. To make this alternative work, we may need to set different learning rates for the scalars and the weights, which adds to the difficulty of hyperparameter tuning. Note we do not learn the scalars when training networks initialized by GradInit.

Table 6: $Acc_1/Acc_{best}$ of ResNet-50 models on ImageNet. Result of MetaInit comes from Dauphin and Schoenholz [16] and we reimplemented the rest.

| | Kaiming | FixUp | MetaInit | GradInit |
|---|---|---|---|---|
| w/ BN | 14.6/75.9 | - | - | 19.2/76.2 |
| w/o BN | - | 18.0/75.7 | -/75.4 | 19.2/75.8 |

**GradInit scales to ImageNet.** As shown in Table 6, GradInit also accelerates convergence and improves test accuracy of ResNet-50 on ImageNet, with or without BN layers, despite having to

use a smaller batch size for GradInit than training due to our GPU memory limit. The acceleration achieved by GradInit is even more significant than FixUp, even on the network with the architecture designed for the initialization.

## 4.2 Training the Original Transformer Model without Warmup

For a Transformer model to converge, either an explicit or implicit learning rate warmup stage is needed, especially for the original Transformer architecture. It is observed that this Post-LN architecture tends to outperform the Pre-LN model [6] while having higher gradient variance at initialization [4]. Is it believed that this high variance makes a warmup stage inevitable. Previous works that removes the warmup stage often involves architectural changes, e.g., removing Layer Normalizations, since it can surprisingly cause instability [4]. Here, we show that with a proper initialization, we can do away with the warmup stage for the original Post-LN Transformer without any modification to the architecture. Table 7 summarizes the architectural changes and best results of methods for improving the initialization of Post-LN Transformers. We compare the stability of the GradInit and Admin initialization methods without warmup in Figure 3.

Table 7: A comparison of GradInit with with the results from the papers (top 4 rows), and our reimplementation of Admin for training the Post-LN Transformer model on the IWSLT-14 De-EN dataset. "Standard" refers to training with standard initialization and warmup.

| Method | Remove LN | $\boldsymbol{w}_{skip}$ | Warmup | Optimizer | BLEU |
|--------|-----------|-------------------------|--------|-----------|------|
| Standard [6] | | | ✓ | RAdam | 35.6 |
| FixUp [9] | ✓ | | ✓ | Adam | 34.5 |
| T-FixUp [5] | ✓ | | | Adam | 35.5 |
| Admin [6] | | ✓ | | RAdam | 35.7 |
| Admin | | ✓ | | Adam | 36.1 |
| Admin | | ✓ | | SGD | 33.7 |
| GradInit | | ✓ | | Adam | 36.0 |
| GradInit | | | | Adam | 36.1 |
| GradInit | | | | SGD | 35.6 |

**Dataset, Architecture, & Hyperparameters.**

IWSLT'14 DE-EN [28] is a German to English translation dataset that has 160k training examples. Our Transformer model is inherited from [3], which is a Post-LN Transformer placing its Layer Normalization after the summation of the skip connection and the residual branch. It has a 512-dimensional word embedding layer and 1024 dimensions in its hidden FFN layer. We also apply GradInit to the variant from Admin [6], where a learnable vector $\boldsymbol{w}_{skip}$ is element-wise multiplied with each dimension of the skip connection, but we initialize it to 1 for GradInit. Please refer to [6] for how Admin initializes these weights. Following [6], we use a linearly decaying learning rate schedule that decays from the maximum learning rate $\eta_{max}$ to 0 as the model trains for 100K iterations. For training with SGD, we set the prescribed learning rate $\eta_{max} = 0.15$, and use $\eta = 0.15, \gamma = 1$ for GradInit. We do a grid search on $\eta_{max}$ for Admin and report its best result in Table 7. For training with Adam, we set $\eta = 5 \times 10^{-4}, \gamma = 10^3$ for the objective of GradInit, so that $\eta\gamma$ is $O(10^{-1})$ as discussed in Section 3.3. We train the initialized model $\eta_{max}$ and $\beta_2$ as listed in Figure 3. We evaluate the BLEU score every epoch, and report the best BLEU scores throughout training for each run. For GradInit, we set the maximum number of iterations $T$ to 780. By comparison, the warmup stage usually takes 4000 iterations, and we find that if we use 780 steps for warmup, the model does not converge with $\eta_{max} \geq 3 \times 10^{-4}$. For $\eta_{max} = 2 \times 10^{-4}$ with 780-step warmup, the BLEU score is 35.4, worse than GradInit's 36.0, showing the advantage of GradInit against warmup.

**Stability after removing warmup for Adam.** In Figure 3, the training process becomes more unstable as $\beta_2$ grows larger. From the analysis of RAdam [35], this is because the variance of the gradient has a stronger impact on the adaptive learning rate when $\beta_2$ is closer to 1. Therefore, the largest $\beta_2 < 1$ that maintains the performance of the trained model reflects the stability of the initialization. We can see GradInit results in more stable models than Admin in general, though their best performance numbers are almost the same. In addition, we find $\boldsymbol{w}_{skip}$ can help stabilize training in extreme hyper parameter settings, e.g., at $\eta_{max} = 5 \times 10^{-4}$ and $\beta_2 = 0.995$ in Figure 3, GradInit with

$\boldsymbol{w}_{skip}$ obtains a good average BLEU score of 36.0, while without $\boldsymbol{w}_{skip}$ only succeeded in obtaining a BLEU score $> 35$ for one out of four experiments, resulting in an average BLEU score of 8.9.

We also find the network is unable to be trained without learning rate warmup if we just fix $\boldsymbol{w}_{skip}$ to its initial value given by Admin and leave the initialization of other parameters unchanged. Nevertheless, with GradInit, we do not need to modify the architecture of Post-LN Transformer to obtain the same good result as Admin. For a closer look at the stabilization mechanism, we show the weight norms and gradient variance at initialization of the original Post-LN architecture using GradInit and Xavier initialization in Figure 9 of the Appendix. For Xavier initialization, the gradient variance is relatively higher for all encoder layers, so GradInit downscales the encoder layer weights more in general. For the LN weights, GradInit only

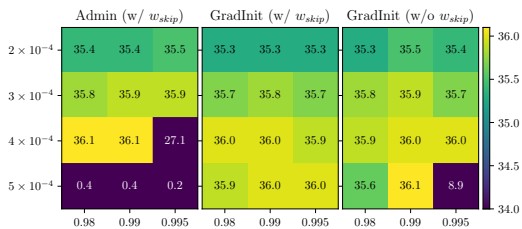

Figure 3: BLEU scores for the Post-LN Transformer without learning rate warmup using Adam on IWSLT-14 DE-EN under different learning rates $\eta_{\max}$ ($y$ axis) and $\beta_2$ ($x$ axis). Each result is averaged over 4 experiments.

downscales the final LN of both the encoder and decoder, which reduces the variance of the encoder and decoder during the forward pass. Another strategy GradInit learns is to downscale the weights of the output projection and the FFN layers, so that the residual branch is relatively down-weighted compared with the skip connection, similar to Admin.

**Removing warmup without architectural change.** Another widely observed phenomenon is that adaptive methods such as Adam seem to be much better than SGD for training Transformer-based language models [13]. Table 7 shows that, with GradInit, we can find a good initialization for the Post-LN Transformer on IWSLT-14 DE-EN that trains using SGD *without learning rate warmup nor gradient clipping*, and achieves performance close to Adam trained using the same type of learning rate schedule. By comparison, Admin also makes the Transformer trainable with SGD, but the BLEU score is lower than the one initialized with GradInit. By comparing Figures 9 and 10 in the Appendix, we find GradInit for Adam and SGD adopts different rescaling patterns, with the Adam version depending more on downscaling the residual branches through the FFN and output projection layers than the SGD version, and the SGD version downscaling more in the final FFN block of the decoder. This highlights the importance of considering the optimization algorithm $\mathcal{A}$ in GradInit, and also indicates the presence of different ways to reduce the initial gradient variance.

## 5   Conclusion

In this paper, we propose *GradInit*, a gradient-based initialization scheme for any architecture. GradInit reinitializes a network by learning a scale factor for each randomly initialized parameter block of a network, so that the training loss evaluated on a different minibatch after one gradient step of a specific stochastic optimizer is minimized. Such a design takes the stochasticity, the learning rate, and the direction of the optimizer into account, allowing us to find better initializations tailored for the optimizer. The initialization learned by GradInit often decreases the gradient variance for most of the parameter blocks. We show that GradInit accelerates the convergence and improves the test performance of a variety of architectures on image classification. It also enables training the Post-LN Transformer without any form of learning rate warmup, even for SGD. GradInit can be a useful tool in the future discovery of better neural architectures that are otherwise discarded due to poor initializations. By analyzing the learned scaling coefficients and their impact on gradient variance, it can also serve a guide to design better initialization schemes for complex architectures to shorten the training schedule and save energy.

## 6   Acknowledgement

This project was supported by the Office of Naval Research, AFOSR MURI program, the DARPA Young Faculty Award, and the National Science Foundation Division of Mathematical Sciences. Additional support was provided by Capital One Bank and JP Morgan Chase.

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
