# GradInit: Learning to Initialize Neural Networksfor for Stable and Efficient Training (Appendix)

## A   Experimental Details

### A.1   On CIFAR-10

**Architectures.**   The base architectures include a popular variant of VGG-19 [36] with BN for CIFAR-10, which includes all the sixteen convolutional layers but only one fully connected layer; a ResNet-110 [21] with base width 16 and two Conv layers in each residual block, as well as its 1202-layer verison with the same depth configurations as FixUp; a 28-layer Wide ResNet [37] with Widen Factor 10 (WRN-28-10) ; and a DenseNet-100 [38]. To isolate the effect of BN, we also consider removing the BN layers from these three networks and adding learnable bias parameters in their place. To compare with a strong initialization scheme that is tailor-made for an architecture family, we consider a 110-layer FixUpResNet [9]. FixUpResNet removes the BN from ResNet, replacing it with bias parameters and a learnable scale parameter after the second convolutional layer of each block. FixUp initializes the weights of the second convolutional layer in each residual block, and of the final fully connected layer, to zero. It also scales the first convolutional layer in each residual block by $1/\sqrt{M}$. This causes the gradient to be zero in the first step for all layers except for the final FC layer. When testing GradInit on this architecture, we adopt the non-zero Kaiming initialization to all convolutional and FC layers. The results are given in Table 10.

**Additional Training Hyerparameters.**   We use batch size 128 to train all models, except for DenseNet-100, where the recommended batch size is 64.[2] We use random cropping, random flipping and cutout [39] for data augmentation. We do not use dropout in any of our experiments. We set weight decay to $10^{-4}$ in all cases.

**Configurations for GradInit.**   As in Algorithm 1, each scale factor is initialized to 1 and we set lower bounds $\alpha = 0.01$. For each architecture, we try $\tau$ from $\{10^{-3}, 2 \times 10^{-3}, 5 \times 10^{-3}, 10^{-2}, 2 \times 10^{-2}, 5 \times 10^{-2}, 10^{-1}\}$, and report the results of 4 runs with the best $\tau$. We find the best $\tau$ for VGG-19 (w/o BN), VGG-19 (w/ BN), ResNet-110 (w/o BN), ResNet-110 (w/ BN), FixUpResNet, DenseNet-100 (w/o BN), DenseNet-100 (w/ BN) are $10^{-2}, 10^{-1}, 5 \times 10^{-2}, 5 \times 10^{-3}, 2 \times 10^{-2}, 5 \times 10^{-3}, 10^{-2}$ respectively.

### A.2   On ImageNet

We use random cropping and flipping as data augmentation. For experiments without BN, we additionally apply MixUp [40] with $\alpha = 0.7$ for all models, for fair comparisons FixUp. We train the models for 90 epochs and decay the learning rate by a factor of 10 every 30 epochs. To fit into the memory, we use a batch size of 128 for GradInit.

We simply run GradInit for 2000 iterations, which is less than half an epoch. Considering ImageNet and CIFAR-10 has 1000 and 10 classes respectively, the cross entropy loss of a random guess on ImageNet is 3 times as large as the loss on CIFAR-10, so a proper initial gradient norm for ImageNet may be 3 times as large as that for CIFAR-10. Therefore, we set $\gamma = 3$ for ImageNet. Since $\tau = 10^{-2}$ worked the best for ResNet-110 (w/ BN) on CIFAR-10, we tried $\tau \in \{1 \times 10^{-3}, 3 \times 10^{-3}, 5 \times 10^{-3}, 10^{-2}\}$ on ImageNet, and chose $\tau = 3 \times 10^{-3}$, which maximizes the test accuracy of first epoch.

### A.3   On Machine Translation

For training with SGD, we fix the momentum to 0.9, and did a grid search fo the prescribed learning rate $\eta_{\max}$ from 0.05 to 0.2 just to present its best result. During this grid search process, we set the $\eta$ of GradInit to $\eta = \eta_{\max}$ We find using $\eta_{\max} = 0.15$ gives the best results, though the model with $\eta_{\max}$ obtained a similar BLEU of 35.4. We also set $\eta = 0.15, \gamma = 1$ for GradInit in this case. We did

---

[2] https://github.com/gpleiss/efficient_densenet_pytorch

a grid search on learning rates from $\{0.01, 0.025, 0.05, 0.06, 0.07, 0.08, 0.09, 0.1\}$ for Admin. We find it achieves the best result with learning rate 0.06, and diverges when $\eta_{\max} > 0.06$. For training with Adam, we set $\eta = 5 \times 10^{-4}$ for the objective of GradInit, and tried $\eta_{\max}$ and $\beta_2$ as listed in Figure 3. We evaluate the BLEU score every epoch, and report the best BLEU scores throughout training for each method. For GradInit, we set the maximum number of iterations $T$ to 780. By comparison, the warmup stage usually takes 4000 iterations. As discussed in Section 3.3, we also set $\gamma = 10^3$.

# B    Mini-batching, continued: Choice of $\tilde{S}$

| Model (#Params) | $r = \lvert \tilde{S} \cap S \rvert / \lvert S \rvert$ | $\tau$ | $\lVert \boldsymbol{g} \rVert_2$ | $Acc_0$ | $Acc_{best}$ |
|---|---|---|---|---|---|
| VGG-19 | 0.5 | $4 \times 10^{-3}$ | $8.63 \pm 0.20$ | $\mathbf{38.37 \pm 1.45}$ | $\mathbf{94.78 \pm 0.08}$ |
| w/ BN | 1 | $1 \times 10^{-4}$ | $11.56 \pm 0.05$ | $13.81 \pm 2.47$ | $94.45 \pm 0.07$ |
| (20.04 M) | 1 | $4 \times 10^{-3}$ | $190.62 \pm 7.65$ | $10.30 \pm 0.15$ | $93.70 \pm 0.17$ |

Table 8: Using GradInit without the gradient norm constraint with different overlapping ratios $r$ to initialize and train a VGG-19 (w/ BN). For both $r = 0.5$ and $r = 1$, we tried $\tau$ from the range of $1 \times 10^{-4}$ to $2 \times 10^{-2}$. The first two rows show the results with the best final test accuracy $Acc_{best}$ among different $\tau$'s, while the last row shows using a larger $\tau$ for $r = 1$.

The choice of $\tilde{S}$, the minibatch on which the objective $L(\tilde{S}; \boldsymbol{\theta} - \eta \mathcal{A}[g(S; \boldsymbol{g})])$ is evaluated, has great influence on the results. We have chosen $\tilde{S}$ to have 50% of its samples from $S$ to reduce the variance of the objective. If $\tilde{S}$ is a completely new batch, i.e., the overlapping ratio $r = \lvert \tilde{S} \cap S \rvert / \lvert S \rvert$ is 0, then it becomes difficult for GradInit to work with some models with high initial gradient variance. On the other hand, when we use the same minibatch ($\tilde{S} = S$), the objective does not capture the stochasticity of the optimizer $\mathcal{A}$ and can cause undesirable results in some cases. We study the effect of different choices of $\tilde{S}$ through VGG-19 networks on CIFAR-10. We consider two settings.

In the first setting, we evaluate VGG-19 (w/o BN) initialized with GradInit using different overlapping ratios of $S$ and $\tilde{S}$. The results are given in Table 8. As we have shown in Figure 6, VGG-19 (w/o BN) has high initial gradient variance. When $r = 0$, sometimes the test accuracy after the first epoch is only 10%, which is worse than the baseline without GradInit. This indicates when $r = 0$, the high variance of $L(\tilde{S}; \boldsymbol{\theta} - \eta \mathcal{A}[g(S; \boldsymbol{g})])$ hinders the effectiveness of GradInit. When $r = 1$, GradInit does effectively reduce the initial gradient variance, achieving lower variance in the first-epoch test accuracy ($Acc_0$) and higher final test accuracy ($A_{test}$) than the baseline (Kaiming Initialization in Table 3), but the result is not as ideal as using $r = 0.5$. We leave more fine-grained evaluation on the choice of overlapping ratio $r$ as future work.

In the second setting, we consider removing the gradient norm constraint of GradInit (by setting $\gamma$ to $\infty$) while using overlapping ratios $r = 1$ and $r = 0.5$ respectively for a VGG-19 (w/ BN) model. We remove the gradient norm constraint to highlight the different degrees of reliance of the two approaches on the gradient constraint. As shown in Table 8, when $r = 1$, we have to use the smallest $\tau$, which results in minimum change to the scale factors, to obtain results that are not significantly worse than the baseline (Kaiming initialization listed in Table 3). It is easy for these large over-parameterized models to overfit a single minibatch with the scale factors. When $r = 1$, GradInit learns a greedy strategy, which increases the gradient as much as possible to enable a steeper descent that sometimes can reduce the loss on the same minibatch by more than 50% in just one iteration. The greedy strategy tends to blow up of the gradient norm at initialization, which hinders convergence and results in a higher dependence on the gradient norm constraint $\gamma$. However, when we use $\tau = 0.5$, GradInit is able to improve the baseline without any gradient norm constraint.

# C    Magnification Effect of BN

Intuitively, if we stop the gradient passing through the mean and bias of the BN layer during backpropagation, the BN layer will magnify the gradient variance when the variance of its input features is smaller than 1 in the forward pass. Here we show its magnification effect analytically for the practical case where the gradient is not stopped for the mean and bias of the BN layer. From

the input to the output, the layers are usually ordered as Linear, BN, Activation. Without loss of generality, we assume the linear layer before BN is $X = ZW + b$, where the output features $X = [x_1, ..., x_n]^T \in \mathbb{R}^{n \times d}$, the input activations $Z \in \mathbb{R}^{n \times k}$, $n$ is the number of samples and $d, k$ are the dimension of each feature vector. Batch Normalization normalizes each activation vector $x_i$ as following

$$y_i = \gamma \frac{x_i - \mu}{\sqrt{\sigma^2 + \epsilon}} + \beta, \tag{3}$$

where all operators are element-wise, $\gamma, \beta \in \mathbb{R}^d$ are learnable parameters usually initialized to 1 and 0 respectively, $\epsilon > 0$ is a small constant for numerical stability, and

$$\sigma^2 = \frac{1}{n} \sum_{i=1}^{n} (x_i - \mu)^2, \mu = \frac{1}{n} \sum_{i=1}^{n} x_i. \tag{4}$$

For most initialization schemes, $b$ is initialized to 0. $\epsilon$ is often small and ignorable. Under these two assumptions, each $y_i$ is invariant to the rescaling of $W$. Rescaling $W$ changes the scale of $x_i$, $\sigma$ and $\mu$ homogeneously. Therefore, among all the parameters of the network, if we only change $W$ by rescaling it into $\alpha W$ ($\alpha > 0$), then $y_i$ does not change, and consequently, $\text{Var}[y_i]$, $\frac{\partial L}{\partial y_i}$ and $\text{Var}[\frac{\partial L}{\partial y_i}]$ do not change, but $\sigma^2$ becomes $\alpha^2 \sigma^2$. To see the magnification effect on the gradient variance during backward propagation, we first find the relation between $\frac{\partial L}{\partial y_i}$ and $\frac{\partial L}{\partial (\alpha x_i)}$ under different scales $\alpha$. In fact,

$$\frac{\partial L}{\partial (\alpha x_i)} = \frac{\gamma}{n\sqrt{\alpha^2 \sigma^2 + \epsilon}} \left[ n \frac{\partial L}{\partial y_i} - \sum_{j=1}^{n} \frac{\partial L}{\partial y_j} - \frac{y_i - \beta}{\gamma} \sum_{j=1}^{n} \frac{\partial L}{\partial y_j} \cdot \frac{y_j - \beta}{\gamma} \right], \tag{5}$$

where, again, all operations are element-wise. Therefore, when $\alpha$ is larger, the variance of the input feature $\alpha^2 \sigma^2$ is larger, and the gradient variance becomes smaller after propagated through this BN layer. Since $Z$ remains the same, $\text{Var}\left[\frac{\partial L}{\partial W}\right]$ becomes smaller. This explains why GradInit learns to enlarge the weights of Conv layers in the VGG-19 (w/ BN) experiments. Further, to simplify the analysis and show its magnification effect on gradient variance when $\alpha^2 \sigma^2 < 1$, let $\gamma = 1, \beta = 0$, and we assume each dimension of $\frac{\partial L}{\partial y_i}$ is i.i.d., and $y_i$ is independent from $\frac{\partial L}{\partial y_i}$, which is not necessarily a stronger assumption than [1, 2], then

$$
\begin{aligned}
\text{Var}\left[\frac{\partial L}{\partial (\alpha x_i)}\right] &= \frac{1}{n^2(\alpha^2 \sigma^2 + \epsilon)} \text{Var}\left[ n \frac{\partial L}{\partial y_i} - \sum_{j=1}^{n} \frac{\partial L}{\partial y_j} - y_i \sum_{j=1}^{n} \frac{\partial L}{\partial y_j} \cdot y_j \right] \\
&= \frac{1}{n^2(\alpha^2 \sigma^2 + \epsilon)} \text{Var}\left[ (n - 1 - y_i^2) \frac{\partial L}{\partial y_i} - \sum_{j=1, j \neq i}^{n} (1 + y_i y_j) \frac{\partial L}{\partial y_j} \right] \\
&\geq \frac{1}{n^2(\alpha^2 \sigma^2 + \epsilon)} \left\{ (n-1)^2 \text{Var}\left[\frac{\partial L}{\partial y_i}\right] + \sum_{j=1, j \neq i}^{n} \text{Var}\left[\frac{\partial L}{\partial y_j}\right] \right\} \\
&= \frac{n(n-1)}{n^2(\alpha^2 \sigma^2 + \epsilon)} \text{Var}\left[\frac{\partial L}{\partial y_i}\right],
\end{aligned}
\tag{6}
$$

where the inequality comes from the assumption that $y_i$ is independent from $\frac{\partial L}{\partial y_i}$ and the fact that $\text{Var}[(X + a)Y] \geq \text{Var}[X] + a^2 \text{Var}[Y]$ ($a$ is a constant) when $X, Y$ are independent, and the last equality comes from the i.i.d. assumption. Therefore, if $\epsilon$ is ignorable and $\alpha^2 \sigma^2 < \frac{n(n-1)}{n^2}$, we will have

$$\text{Var}\left[\frac{\partial L}{\partial (\alpha x_i)}\right] > \text{Var}\left[\frac{\partial L}{\partial y_i}\right], \tag{7}$$

i.e., the BN layer magnifies the gradient variance when $\alpha^2 \sigma^2$ is small.

## D  Improved Implementation of MetaInit

MetaInit was originally designed to be task-agnostic, and learns to initialize the network with random samples as inputs. Here, for fair comparisons, we also feed training data to MetaInit, as this should

intuitively improve MetaInit for the specific task, and use Adam with the same gradient clipping to optimize the weight norms for MetaInit. Originally, MetaInit [16] uses signSGD with momentum [41], but we found using Adam with the hyperparameters above can give better results for MetaInit. Table 9 shows the comparison before and after the changes.

Table 9: $Acc_1$, $Acc_{best}$ for different versions of MetaInit (4 runs). "rand.": using random data. "real": using real data.

| config | vgg19 w/o BN | vgg19 w/ BN | res.110 w/o BN | res.110 w/ BN |
|---|---|---|---|---|
| rand. + signSGD | 29.08, 94.36 | 15.62, 94.53 | 15.91, 93.91 | 24.47, 94.93 |
| real + signSGD | 30.89, 94.41 | 16.58, 94.46 | 16.21, 94.29 | 26.28, 94.95 |
| real + Adam | 30.48, 94.62 | 35.09, 94.64 | 14.55, 94.19 | 29.00, 94.76 |

# E    Additional Experimental Results

We give additional results on WRN-28-10, FixUpResNet and DensetNet-100 on CIFAR-10 in Table 10.

Table 10: First epoch ($Acc_1$) and best test accuracy over all epochs ($Acc_{best}$) for models on CIFAR-10. We report the mean and standard error of the test accuracies in 4 experiments with different random seeds. Best results in each group are in bold. For WRN, we have additionally used MixUp during training to enhance the results, but we do not consider mixup for GradInit to test its transferability to different training regularizations. Its result with MetaInit comes from the MetaInit paper.

| Model (# Params) | | WRN-28-10 w/ BN (36.49M) | FixUpResNet N/A (1.72M) | DenseNet-100 w/o BN (0.75M) | DenseNet-100 w/ BN (0.77M) |
|---|---|---|---|---|---|
| Kaiming | $Acc_1$ | $43.1 \pm 2.7$ | $38.2 \pm 0.8$ | $35.5 \pm 0.6$ | $51.2 \pm 1.5$ |
| | $Acc_{best}$ | $97.2 \pm 0.1$ | $\mathbf{95.4} \pm 0.1$ | $94.0 \pm 0.1$ | $95.5 \pm 0.1$ |
| MetaInit | $Acc_1$ | - | $21.5 \pm 0.6$ | $35.1 \pm 0.2$ | $46.7 \pm 4.0$ |
| | $Acc_{best}$ | $97.1$ | $95.0 \pm 0.1$ | $94.4 \pm 0.1$ | $95.5 \pm 0.1$ |
| GradInit | $Acc_1$ | $46.3 \pm 0.4$ | $35.0 \pm 0.7$ | $37.2 \pm 1.1$ | $58.2 \pm 0.9$ |
| | $Acc_{best}$ | $\mathbf{97.3} \pm 0.1$ | $\mathbf{95.4} \pm 0.1$ | $\mathbf{94.9} \pm 0.1$ | $95.5 \pm 0.1$ |

# F    Weight norms and gradient variances

In this section, we give weight norms and gradient variances before and after GradInit is applied on various datasets and networks. We consider DenseNet-100 (w/o BN) and DenseNet-100 (w/ BN) on CIFAR-10 in Figure 7 and Figure 8, as well as ResNet-50 on ImageNet in Figure 2. We also compare the weight norms and gradient variances of the Post-LN Transformer model initialized using GradInit with $\mathcal{A}$ set to Adam and SGD respectively in Figure 9 and Figure 10.

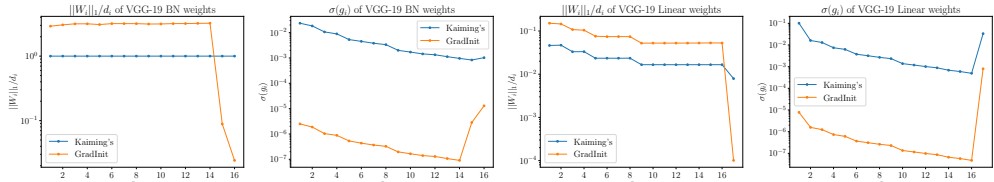

Figure 4: Averaged per-dimension weight magnitudes ($\|W_i\|/d_i$) and standard deviation of their gradient ($\sigma(\boldsymbol{g}_i)$) for each layer $i$ of the VGG-19 (w/ BN) on CIFAR-10. The ratio between the weight magnitudes of GradInit and Kaiming Initialization is the learned scale factor of GradInit in each layer. The standard deviation is computed over the minibatches, with a batch size of 128, with the BN in its training mode. This VGG-19 on CIFAR-10 has only one FC layer, but it has the same number of convolutional layers (16) as its ImageNet version. All the weights are indexed from shallow to deep, so the first 16 entries of the Linear Weights are of Conv layers, while the 17th is the FC layer. Due to the magnification effect of BN, $\sigma(\boldsymbol{g}_1)/\sigma(\boldsymbol{g}_{16})$ for the Conv layers is higher than it is in VGG-19 without BN, shown in Figure 6. GradInit learns to reduce the magnification effect of BN layers by scaling up all the Conv layers and most of the BN layers, given it has greatly down scaled the last two BN layers and the final FC layer to reduce the variance in the forward pass.

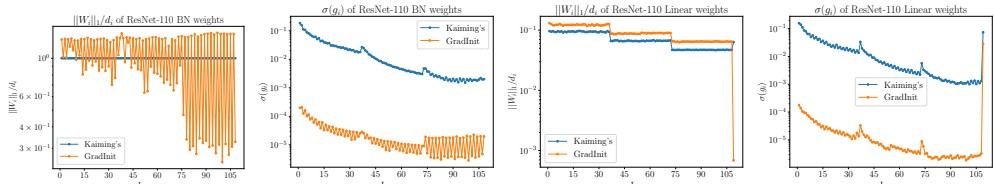

Figure 5: Averaged per-dimension weight magnitude ($\|W_i\|/d_i$) and standard deviation of their gradient ($(\sigma(\boldsymbol{g}_i))$) of the Batch Normalization (BN) layers and the linear layers of the ResNet-110 on CIFAR-10. All the layers are indexed from shallow to deep. The linear layers include all Conv layers (2 for each of the residual blocks) and the final FC layer. The ratio between the weight magnitudes of GradInit and Kaiming Initialization is the learned scale factor of GradInit in each layer. The gradient variance is computed with a batch size of 128. GradInit finds a combination of weight norms where the gradient variance is reduced for all layers. Specifically, it learns to further scale down the second BN layer of each residual block in deeper layers, which is a useful strategy, as deeper layers should have less marginal utility for the feature representations, and scaling down those layers helps to alleviate the growth in variance in the forward pass [10]. GradInit also learns to scale up weights of the first BN layer and all the Conv layers in each residual block, which alleviates the magnification effect of the BN layers on the gradient variance during backpropagation, happening if their input features in the forward pass have small variances. The jump on the curves occur when the dimension of the convolutional filters changes.

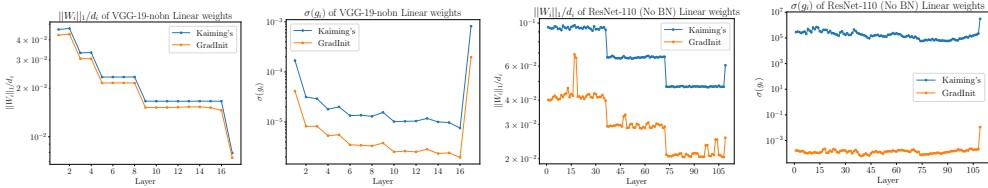

Figure 6: Averaged per-dimension weight magnitude ($\|W_i\|/d_i$) and standard deviation of their gradient ($(\sigma(\boldsymbol{g}_i))$) of the VGG-19 (left two) and ResNet-110 (right two) without BN on CIFAR-10, evaluated with a batch size of 128. For VGG-19 (w/o BN), $\sigma(\boldsymbol{g}_i)$ increases at Conv layers with different input and output dimensions during backpropagation. For ResNet-110 without GradInit, the gradient variance is very high due to the cumulative effect of skip connections during the forward pass. In this scenario, to reduce the gradient variance, there is no reason to increase the weights, so GradInit downscales the weights for all layers in both architectures, unlike the case with BN.

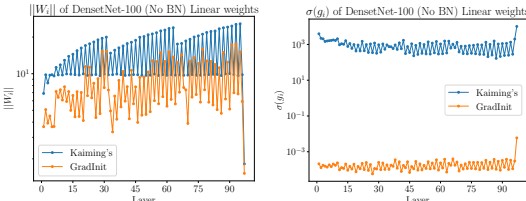

Figure 7: Averaged per-dimension weight magnitudes ($\|W_i\|/d_i$) and standard deviation of their gradient ($\sigma(\boldsymbol{g}_i)$) for each linear layer $i$ in DenseNet-100 (w/o BN). All the layers are indexed from shallow to deep. The linear layers include all convolutional layers and the final fully connected layer. Inside each dense block, each layer concatenates all the preceding features, so their input dimension increases, the weight dimension increases and the weight norm increases. Compared with Figure 6, DenseNet-100 does not significantly increase the gradient variance during backpropagation. The standard deviation of the gradient is reduced by around $10^6$ with GradInit, which explains why it is possible to train DenseNet-100 (w/o BN) without gradient clipping after using GradInit. The major source of gradient reduction of GradInit comes from reducing the weights in each layer.

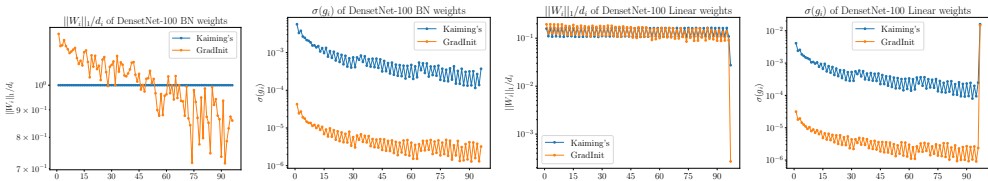

Figure 8: Averaged per-dimension weight magnitudes ($\|W_i\|/d_i$) and standard deviation of their gradient ($\sigma(\boldsymbol{g}_i)$) for each (BN or linear) layer $i$ in the DenseNet-100 (w/ BN). All the layers are indexed from shallow to deep. The linear layers include all convolutional layers and the final fully connected layer. The major source of variance reduction comes from down-scaling the final FC layer.

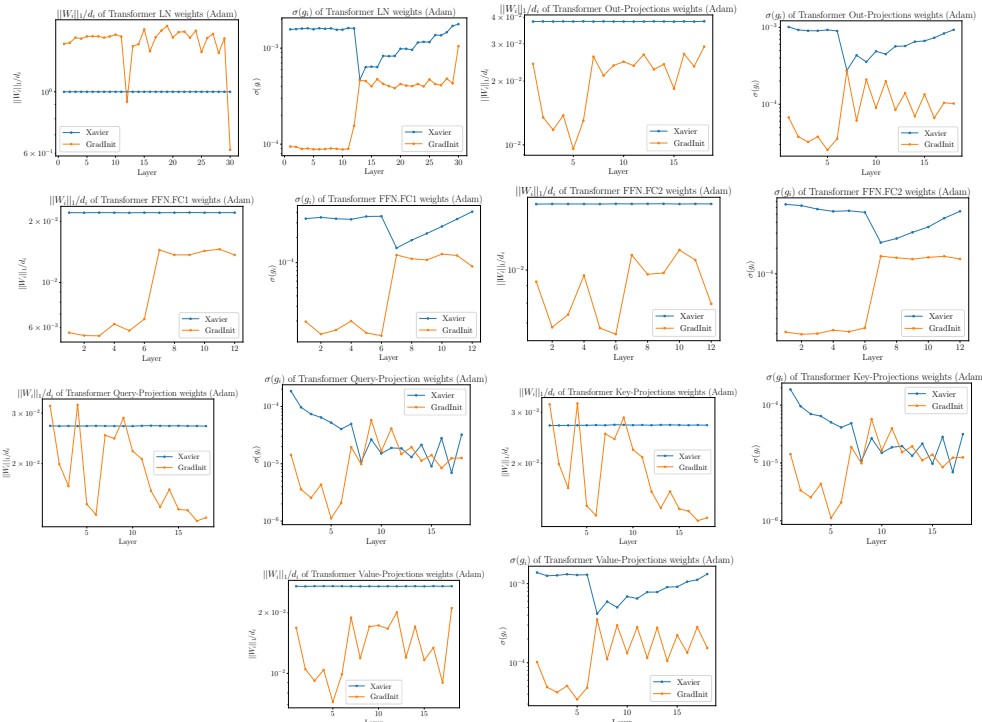

Figure 9: Weight norm and averaged per-dimension standard deviation of each weight of the normalization layers and linear layers in the Post-LN Transformer. Here, GradInit sets $\mathcal{A}$ to Adam. The Transformer has 6 Transformer blocks in its encoder and decoder. In each plot, we first list the values for weights in the encoder, and then those in the decoder. Inside each encoder, we first list the weights from the self attention layers and then the those from the FFN layers. Inside each decoder, we first list the weights in the order of self attention, encoder attention and FFN. In general, GradInit reduces the variance for all the weights, except for some of the Query-Projection and Key-Projection weights in the decoder, which are inside the softmax operations in the self attention blocks. The major source of gradient variance reduction comes from downscaling the final LN weights of the decoder, as well as the linear layers of each residual branch (Out-Projection and Value-Projection weights, FFN.FC1 and FFN.FC2 weights) *in each block*.

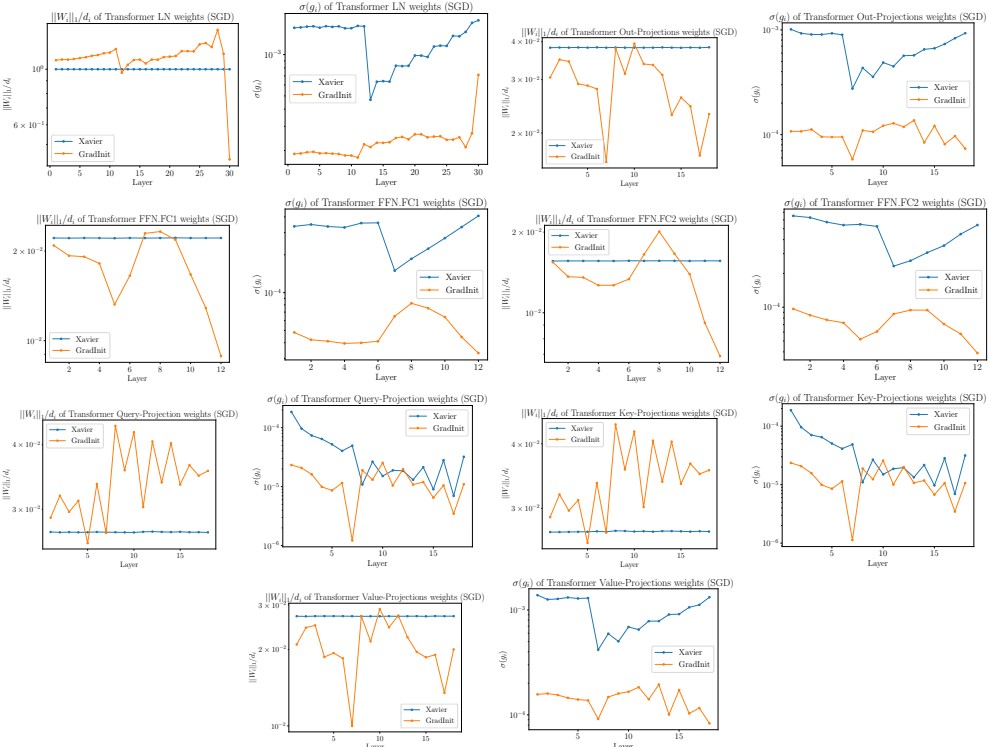

Figure 10: Weight norm and averaged per-dimension standard deviation of each weight of the normalization layers and linear layers in the Post-LN Transformer. Here, GradInit sets $\mathcal{A}$ to SGD. The Transformer model and the way each weight is permuted are the same as in Figure 9. Again, in general, GradInit reduces the variance for most of the weights, except for some of the Query-Projection and Key-Projection weights in the decoder, which are inside the softmax operations in the self attention blocks. Different from the patterns in the Adam version, which downscale all the weights in every layer except for the Query-Projection and Key-Projection weights, the SGD version of GradInit mostly reduces the weights in the final Transformer block of the decoder.