# OpenReview forum: "GradInit: Learning to Initialize Neural Networks for Stable and Efficient Training"
_NeurIPS.cc/2021/Conference — NeurIPS 2021 Poster_

### Official Review · Reviewer_QztT · 2021-07-15

**Rating:** 7
**Confidence:** 3

**Summary:**

The authors introduce a new automated and architecture agnostic method to initialize the weights of a neural network. The method incorporates a pretraining step where a scalar coefficient for each of the weight matrices is optimized to minimize the loss on the first step of the training pipeline, subject to a constrained gradient norm. The authors show that their initialization method achieves better performance than other initialization schemes on a variety of image classification tasks applied to a medley of different architectures and a machine translation task applied to a Post-LN Transformer architecture.

**Ethical Concerns:**

There are no ethical issues with this paper.

**Limitations And Societal Impact:**

The authors have adequately addressed the limitations of their work.

**Main Review:**

Originality:
Other initialization schemes have been proposed in the past such as MetaInit. However, GradInit uses batches of training data and is more computationally efficient as well as memory efficient. It is also more adaptable to different optimizers and hyperparameters.

Quality:
This is a high-quality paper with most of the claims supported through experimental results. However, I would have appreciated more experiments with non-residual architectures. The authors claim that their method generalizes to all kinds of architectures, but in their experiments, they only compare their method on VGG-19, ResNet, Wide ResNet, and DenseNet architectures for CIFAR-10, the ResNet architecture for ImageNet, and the Post-LN Transformer architecture for language. Perhaps the authors could show that their initialization method works for other more exotic architectures from the Neural Architecture Search literature which are proven to have higher performance than ResNet models.

In the Table 4 results, the authors claim that the FixUp $Acc_{best}$ is 75.7, but the FixUp paper claims that they received a final accuracy of 76% for the same experiment. I am not sure why there is a discrepancy here.

Clarity:
Most of the paper is written very clearly. I especially enjoyed the method section where the optimization procedure and the algorithm were invaluable to my understanding of the method. However, there were a few parts of the paper that were not as clear to me:
- How is the grid search for $\tau$ performed? Is there a held-out validation set that $\tau$ is fine-tuned on?
- Table 1 is unclear. I assume the model was consistent across the experiment. However, Table 1 seems to indicate that there are two models: VGG-19 and w/o BN.

The paper also relies heavily on the Appendices. Certain sections do not make sense without reading the Appendices. For instance, the "GradInit further stabilizes feedforward nets with BN" section relies on Figure 3 to be understood.

Significance:
While the paper does show an improvement in results compared to the other considered initialization schemes, the improvements are very marginal (less than a percentage on the other initialization techniques; some of which are much simpler to implement). This questions the merit of using such a complicated initialization process in practice. In the language setting, adding GradInit to Admin does not seem to show any improvement.

On the other hand, it seems like using GradInit has several other appealing properties like increased stability and efficiency which can be used in newer models that might be more unstable than the current ones.

Other miscellaneous items:
- Line 70: how can there be infinite variance, but a finite 2-order moment?
- Line 81: "possbile" should be "possible".
- Line 138: "is that is" should be "is that it is".
- Line 225: "labbeled" should be "labeled".
- Line 245: "decay the learning rate by 10" is not clear.

**Time Spent Reviewing:**

5

---

> ### Author Response · Authors · 2021-08-10
> **Response to Reviewer QztT**
>
> Thank you for acknowledging our contributions! We will rearrange the contents accordingly and correct the typos. We focus on addressing your concerns below.
>
> 1. Difference in the accuracy of FixUp: why the accuracy of FixUp is 75.7 instead of 76.0.
> - Our implementation is based on the code from the authors of FixUp (https://github.com/hongyi-zhang/Fixup). However, as the title says, this repository is also a re-implementation.
> - The difference might also have been caused by some subtle differences in different versions of PyTorch.
> - Our result is the average of 4 experiments with different random seeds. The best one has an accuracy of 75.9. We cannot find from FixUp paper whether they were reporting averaged results or not.
>
> 2. How is the grid search for $\\tau$ performed? Is there a held-out validation set that $\\tau$ is fine-tuned on?
> - We did not use a validation set that is different from the test set. However, the criterion of the grid search is the test accuracy after the first epoch, which reduces the chance of overfitting to the test set since the following training process can still have a huge impact on the final results.
>
> 3. The model in Table 1.
> - Sorry for the confusion. There is only one model, VGG-19 (w/o BN). We will make this table more readable in the next version.
>
> 4. Significance of the results.
> - Comparing with Admin, GradInit removes the warmup stage for training Post-LN Transformer without the need of adding additional learnable scalars to the residual branch. In Figure 2 (right), we show the model without $w\_{skip}$ initialized by GradInit is more robust to hyperparameters than Admin initialization with $w\_{skip}$ (left). This advantage has the potential to help neural architecture search to find better variants that are otherwise discarded due to poor initialization. We leave the application of GradInit in neural architecture search as an important future work.
> - The improvement can be more significant as the network goes deeper, e.g., the case of ResNet-1202.
> - Many of the problems we considered in the current version are standard benchmark problems for which existing models and hyper-parameters have been fine-tuned over years of work to achieve good results, and for this reason we think competing with, and sometimes beating, existing SOTA implementations does carry some weight.
> - We provide the cumulative best test accuracy for VGG-19, ResNet-110 and ResNet-1202 on CIFAR10 (https://ibb.co/2YD3tbx), from which we can see GradInit does accelerates the convergence significantly.

---

> > ### Comment · Reviewer_QztT · 2021-08-31
> > **Response to Authors**
> >
> > Thank you for taking the time to address my questions.
> >
> > - I appreciate that you have tried to replicate the FixUp method to the best of your ability and were unable to reproduce the results they have published.
> > - It is a little concerning to perform hyperparameter tuning on the test set even if it is for only one epoch as there is always a potential for the hyperparameters to fit to some features in the test set.
> > - Thank you for the clarification in Table 1.
> > - I see that GradInit does carry some significance over the other initialization methods. Though, I am still not entirely convinced on how much more significant it is. This is especially true for non-standard architectures and datasets.
> >
> > I will maintain my original score of 7.

---

### Official Review · Reviewer_AaUJ · 2021-07-16

**Rating:** 6
**Confidence:** 2

**Summary:**

This paper proposes GradInit, a method to learn a set of scaling parameters ahead of training to better initialise networks.  Those parameters, learnt using very few batches in the training set,  scale the weights after initializing them with known initialisers like Kaiming or Xavier. The method aims to provide an automated easy to use approach for neural network initialisation that is agnostic to the model architecture and task at hand.

**Limitations And Societal Impact:**

- Limited evaluation of the proposed method.
- No clear advantage over existing techniques.

**Main Review:**

The paper is well written and touches upon an important problem.

I am not very familiar with the line of research investigating the network initialisation problem. Having said this, I would expect a paper that looks into this problem to follow a systematic evaluation evaluation protocol and, as a result, include substantially more results, specially as far as model architectures are concerned. For example, in the paper only two types of architectures ResNet and VGG are evaluated on image classification tasks. I think this is a particularly weak point of this paper specially since the method is framed as an _initialisation scheme for any architecture_.

If this work (or a future version of it) could compare a much wider set of model architectures (maybe tasks too), it would certainly be of very high value for the community at large.

It would be interesting to report the training/validation curves instead of only the accuracy after the first epoch. I am a bit unsure if comparing accuracy after first epoch given a method that effectively “looks” at the training set ahead of the training, is very informative of the method’s real performance (compared to others such as Kaiming).

From the language models experiments, I only find Figure 2 to clearly show the advantage of GradInit. If the range of $\eta_{\max}$ and $\beta_2$ values considered was wider and still showed GradInit being unaffected, that would strengthen the evaluation section.

**Time Spent Reviewing:**

4

---

> ### Author Response · Authors · 2021-08-10
> **Response to Reviewer AaUJ**
>
> Thank you for acknowledging the importance of our work! We appreciate your constructive suggestions. We try to address your concerns below.
>
> 1. More results on different architectures for image classification.
> - VGG and ResNet models represent two fundamental building blocks, the feedforward networks and skip connections, of various convolutional networks. We have also evaluated GradInit on these networks without BN layers, another important building block.
> - We have included some results of DenseNet on CIFAR10 in Appendix E. For convenience, we list the results in the table below.  For DenseNet w/o BN, the result of GradInit is achieved without gradient clipping, but the baseline (Kaiming initialization) requires clipping to converge.
>
> |                       |             Kaiming             |             MetaInit            |             GradInit            |
> |-----------------------|:-------------------------------:|:-------------------------------:|:-------------------------------:|
> | Model                 |         $Acc_0 / Acc_{best}$        |         $Acc_{0} / Acc_{best}$        |         $Acc_0 / Acc_{best}$        |
> | DenseNet-100 (w/o BN) | 35.5 $\\pm$ 0.6 / 94.0 $\\pm$ 0.1 | 35.1 $\\pm$ 0.2 / 94.4 $\\pm$ 0.1 | 37.2 $\\pm$ 1.1 / 94.9 $\\pm$ 0.1 |
> | DenseNet-100 (w/ BN)  | 51.2 $\\pm$ 1.5 / 95.5 $\\pm$ 0.1 | 46.7 $\\pm$ 4.0 / 95.5 $\\pm$ 0.1 | 58.2 $\\pm$ 0.9 / 95.5 $\\pm$ 0.1 |
>
> - GradInit can be easily applied to other vision tasks. We believe this is an important future work.
>
> 2. Train/validation curves.
> - We plot the curves of cumulative best accuracy at each epoch ($Acc^\*_n = max\_{i<=n} Acc\_i$) for VGG-19, ResNet-110 and ResNet-1202 on CIFAR10. The results are per-epoch average of 4 experiments with different random seeds. You can find the plot at https://ibb.co/2YD3tbx. With GradInit, the models do converge significantly faster.
>
> 3. Advantage over existing methods.
> - Many of the problems we considered are standard benchmark problems for which existing models and hyper-parameters have been fine-tuned over years of work to achieve good results, and for this reason we think competing with, and sometimes beating, existing SOTA implementations does carry some weight.
> - The benefit of GradInit becomes more evident as the network goes deeper, e.g., we achieved  more significant improvements on ResNet-1202 than on ResNet-110.
> - For Post-LN Transformer, GradInit successfully removed warmup and achieved comparable results without changing the architecture, unlike Admin, which has to add learnable weights to the skip connections. This property may, e.g., help neural architecture search to find better variants that are otherwise discarded due to poor initialization.

---

### Official Review · Reviewer_Ztx8 · 2021-07-17

**Rating:** 6
**Confidence:** 4

**Summary:**

This research proposes a novel initialization method for architecture agnostic networks by learning a scalar coefficient of a random weight tensor. Compared with the existing initialization method MetaInit, the proposed method is more computationally efficient and can be directly combined with different optimizers.
The motivation and heuristic of this paper is simple but interesting. However, the reviewer thinks this idea is worth digging deeper and requires more validations, not limited to an initialization method.

########################################################################################

The authors have addressed most of my comments and concerns.
However, the reviewer still thinks that the scope of this paper is a bit narrow, if the scope of the current paper is limited to improving the initialization, the authors should provide some theoretical proof and analysis to enrich the paper.
Therefore the reviewer still keeps the score.


**Limitations And Societal Impact:**

The motivation and heuristic of this paper is simple but interesting. However, the reviewer thinks this idea is worth digging deeper and requires more validations, not limited to an initialization method.

**Main Review:**

(1) In line 38, is “the variance” right? I think it should be the learned scalar coefficient. Please check it.

(2) The advantages of the proposed method mainly lie in that it is independent of the network architectures. And it can be combined with network architecture search in future works. In ideal cases, the proposed method should generate a proper initialization for arbitrary networks. The authors also claimed that other architecture-specific initialization schemes are not always portable
to new architectures. Therefore the effectiveness of the proposed method on networks with more complex architectures, like DenseNet, GoogleNet, should be further validated.

(3) In line 291, you claim that the acceleration achieved by GradInit is even more significant than FixUp. Can you report the specific results including the comparisons of initialization optimization time, training time, training epochs between FixUp and your method? Because your method uses a smaller batchsize and this will cause a longer time for a single epoch in the practical training experiment though you may need fewer epochs to train the model.

(4) The main concern of the reviewer is that this idea is worth digging deeper. For example, considering the motivation of this paper “the norm of each network layer is adjusted so that a single step of SGD or Adam with prescribed hyperparameters results in the smallest possible loss value”, a more straightforward and intuitive solution is to design a new linear layer with scale alpha and bias beta (different from the *trainable* linear layer in batch normalization). This linear layer will be adjusted according to the motivation dynamically during the training process, not limited to the initialization.


**Time Spent Reviewing:**

5 hours

---

> ### Author Response · Authors · 2021-08-10
> **Response to Reviewer Ztx8**
>
> Thank you for acknowledging our contributions and starting the interesting discussions. We focus on addressing your concerns below.
>
> 1. In line 38, is “the variance” right?
> - Thank you for finding this typo! As you said, it should be the scalar coefficient. We will correct this in the next version.
>
> 2. Validate the effectiveness of the proposed method on networks with more complex architectures.
> - Note that we have already verified the effectiveness of our method for VGG and ResNet, with or without BN, on image classification tasks. Our method works well for ResNet up to 1202 layers on CIFAR10. Further, we have also validated its effectiveness on Transformers for Machine Translation. The architecture of Transformer is not less complex than DenseNet or GoogleNet.
> - We have already included the results on DenseNet in Appendix E. For convenience, we copy the results as below. For DenseNet w/o BN, the result of GradInit is achieved without gradient clipping, but the baseline (Kaiming initialization) requires clipping to converge. We plan to add results on GoogleNet in the next version.
> |                       |             Kaiming             |             MetaInit            |             GradInit            |
> |-----------------------|:-------------------------------:|:-------------------------------:|:-------------------------------:|
> | Model                 |         $Acc\_0 / Acc\_{best}$        |         $Acc_0 / Acc_{best}$        |         $Acc_0 / Acc_{best}$        |
> | DenseNet-100 (w/o BN) | 35.5 $\\pm$ 0.6 / 94.0 $\\pm$ 0.1 | 35.1 $\\pm$ 0.2 / 94.4 $\\pm$ 0.1 | 37.2 $\\pm$ 1.1 / 94.9 $\\pm$ 0.1 |
> | DenseNet-100 (w/ BN)  | 51.2 $\\pm$ 1.5 / 95.5 $\\pm$ 0.1 | 46.7 $\\pm$ 4.0 / 95.5 $\\pm$ 0.1 | 58.2 $\\pm$ 0.9 / 95.5 $\\pm$ 0.1 |
>
> 3. Can you report the specific results including the comparisons of initialization optimization time, training time, training epochs between FixUp and your method?
> - In line 291, the smaller batch size is only used during the optimization of the scale factors (Algorithm 1). For the comparisons with FixUp, after 1000 steps of Algorithm 1 (T=1000), we train the network with the same hyperparameters as FixUp for 100 epochs. **Therefore, both GradInit and FixUp takes around 52 hours to train on 4 2080Ti GPUs. GradInit takes an extra ~11 minutes to execute Algorithm 1 for 1000 steps.**
> - A plot of the cumulative best test accuracies at each epoch is given at https://ibb.co/StRKS3h. The improvement may not look as significant as expected; we will rephrase the sentence. However, we believe it is nontrivial for GradInit to achieve improvements over FixUp based on its specifically designed architecture.
> - FYI, we also plotted the cumulative best test accuracies at each epoch on CIFAR10 at https://ibb.co/2YD3tbx, where the improvement is much more significant.
>
> 4. ..a more straightforward and intuitive solution is to design a new linear layer with scale alpha and bias beta (different from the trainable linear layer in batch normalization). This linear layer will be adjusted according to the motivation dynamically during the training process, not limited to the initialization.
> - Thank you for the insightful suggestion! Indeed, this idea is worth further explorations in a separate work. The scope of the current paper is limited to improving the initialization of deep (e.g., ResNet1202) or poorly initialized networks (e.g., Post-LN Transformers) so that they can be trained efficiently. How to efficiently and effectively incorporate the norm constraint and the objective of GradInit into the training process is an important future work.

---

### Official Review · Reviewer_Qphs · 2021-07-19

**Rating:** 7
**Confidence:** 4

**Summary:**

This paper proposes a method to scale the initial parameter blocks by a set of scalar values such as to achieve better optimization properties during the course of training. This paper follows up on previous approaches of similar nature but does it more efficiently and in a simpler way, achieving better results. The paper is clearly written, and it was a pleasure to read.




**Limitations And Societal Impact:**

yes

**Main Review:**

This paper proposes a method to scale the initial parameter blocks by a set of scalar values such as to achieve better optimization properties during the course of training. This paper follows up on previous approaches of similar nature but does it more efficiently and in a simpler way, achieving better results. The paper is clearly written, and it was a pleasure to read.

Overall, this is an important area and this paper achieves good initialization with only 1% of the total compute cost. The paper is also written in a way to convey both intuition as well as good detailed implementation details.
One area where the paper can potentially improve would be to improve the overall accuracies compared to previous methods.

Questions:
- For the section starting with Line 153, it is mentioned that if S and \tilde{S} were the same, not enough randomness would be captured. However, is it not possible to keep S and \tilde{S} the same for the same t, and S can be different for t+1? (i.e. the step t from line 3 of Alg. 1)

- Line 189 mentions the added complexity of 2nd order gradients, however table 2 doesn't show significant increase in running time, so whether or not there are 2nd order derivative terms for penalty method is irrelevant?

- Line 230: Why was only SGD used and Adam not used for the image datasets? what happens to the results if Adam were to be used instead of SGD?

- Typo on line 108
- Minor, but it would be good to discuss Admin before line 303.



**Time Spent Reviewing:**

2.5

---

> ### Author Response · Authors · 2021-08-10
> **Response to Reviewer Qphs**
>
> Thank you for your comments! We will rearrange the content and correct the typos accordingly. We focus on your questions below.
>
> 1. Is the possible to keep $S$ and $\\tilde{S}$ the same for the same t, and use different $S$ for t+1?
> - Yes it is possible. In this way, we simply set $\\tilde{S}\_t=S\_t$ in line 9 of Algorithm 1, and sample a new batch $S_t$ in each iteration in Line 4.
> - However, as we have mentioned in line 164, this will lead to large initial gradients that can destabilize training if we do not enforce the constraint in Eq. 1, and negatively affect the performance even when the constraint is enforced with Algorithm 1, since it triggers more steps to minimize the gradient norms instead of the objective (Eq. 1) due to the tendency of increasing the gradient. Please see the table below for an example on VGG-19 without BN. Overlapping ratio is defined as $\\frac{|S\cap \tilde{S}|}{|\tilde{S}|}$. This setting corresponds to the case where “Overlapping Ratio” is 1 in the table.
> | Overlapping Ratio            | First-epoch Acc. |  Best Acc.         |
> | ------------                          |     --------------       |       ------------     |
> |     0                                   |  21.9$\pm$ 4.4 |   94.5 $\pm$ 0.1 |
> |     0.5                                |  29.3$\pm$ 0.6 |   94.7 $\pm$ 0.1 |
> |     1                                   |  28.7$\pm$ 0.9 |   94.5 $\pm$ 0.1 |
>
> 2. In Table 2, is the computational overhead of 2nd order derivative terms in penalty methods significant?
> -  The overhead is not negligible but also not huge compared to the standard gradient update. When the second order term is active, the relative increase in time consumption is 46.4%, 61.3%, 64.1% and 38% respectively.
> - In addition, Table 2 also shows that the penalty methods require more tuning for the regularization strength $\\lambda$, since the optimal values for different architectures are different. By comparison, in the constrained form, a single $\\gamma$ transfers well across architectures.
>
> 3. Why was only SGD used and Adam not used for the image datasets? what happens to the results if Adam were to be used instead of SGD?
> - We choose SGD since it usually achieves the best results for training convolutional networks on image datasets, so that the baseline results are strong and well-established. In general, using Adam to train CNNs on CIFAR10 results in worse accuracy than using carefully tuned SGD (unless large $\\epsilon$ is used and adaptivity is reduced; e.g., [a] uses $\\epsilon=9475$ on ImageNet). The exact reason behind this seems to be an open problem, but it is likely that SGD has better inductive bias for CNNs on image data.  We chose not to focus on this issue since there is not an established state of the art baseline for Adam training.
> - Our preliminary experiments show that GradInit still improves the convergence with Adam. Without further tuning, we use learning rate 1e-3 and weight decay 0.01 for AdamW (AdamW has better compatibility with weight decay regularization), and use the cosine learning rate schedule to train a VGG-19 (w/ BN) on CIFAR10. With Kaiming Initialization and GradInit, the $(Acc\_0, Acc\_{best})$ is $(39.4 \pm 2.4, 94.0 \pm 0.1)$ and  $(46.5 \pm 0.6, 94.1 \pm 0.1)$ respectively. We plan to add more comprehensive results in this setting in the future.
>
> [a] Choi, D., Shallue, C. J., Nado, Z., Lee, J., Maddison, C. J., & Dahl, G. E. On empirical comparisons of optimizers for deep learning. arXiv preprint arXiv:1910.05446.

---

### Decision · Program_Chairs · 2021-09-28

**Decision:**

Accept (Poster)

**Comment:**

All reviewers were positive about this paper: it suggests an initialization procedure GradInit which improves upon a previous method MetaInit, in accuracy, but especially in terms of computational efficiency. A recurring and remaining concern from the reviews was that the authors claim an "architecture agnostic method", but don't show it in many architectures. I therefore recommend the authors to tone down this claim. Also, I was not sure about the significance of the result that GradInit enables training without learning rate warmup in Post-LN Transformer, since it is not clear to me if warm-up is necessarily more expensive than running GradInit. Therefore, I recommend that the authors should mention this in the final version.

**Consistency Experiment:**

NeurIPS has a long history of experimentation. In 2014, NeurIPS ran an experiment in which 10% of submissions were reviewed by two independent committees to quantify the randomness in the review process. This year, we repeated a variant of this experiment to see how the quality of the review process has changed over time.  This paper was part of the experiment and was therefore assigned to two committees (consisting of reviewers, an Area Chair, and a Senior Area Chair) that reached independent decisions.  If both committees made the same recommendation, this recommendation was followed. If a single committee recommended acceptance, the paper was accepted (with the exception of a few cases in which the other committee identified what we considered a fatal flaw, e.g., an error in a key result).

Both committees reached the same decision: **Accept (Poster)**

The other committee assigned to the paper recommended **Accept (Poster)**.  You can find the other set of reviews, along with any follow up discussion with the authors here:
https://openreview.net/forum?id=eXlxB3aLOe